# Tablets as an Option for Telemedicine—Evaluation of Diagnostic Performance and Efficiency in Intracranial Arterial Aneurysm Detection

**DOI:** 10.3390/diagnostics12102461

**Published:** 2022-10-11

**Authors:** Elif Can, Pimrapat Gebert, Elisa Birgit Sodemann, Johannes Kolck, Thula Canon Walter-Rittel, Anna Maaßen, Christopher Güttler, Juliane Stöckel, Georg Bohner, Georg Böning

**Affiliations:** 1Department of Radiology, Charité—Universitätsmedizin Berlin, Corporate Member of Freie Universität Berlin, Humboldt-Universität zu Berlin, and Berlin Institute of Health, Augustenburger Platz 1, 13353 Berlin, Germany; 2Institute of Biometry and Clinical Epidemiology, Charité—Universitätsmedizin Berlin, Corporate Member of Freie Universität Berlin, Humboldt-Universität zu Berlin, Charitéplatz 1, 10117 Berlin, Germany; 3Department of Nuclear Medicine, Charité—Universitätsmedizin Berlin, Corporate Member of Freie Universität Berlin, Humboldt-Universität zu Berlin, and Berlin Institute of Health, Augustenburger Platz 1, 13353 Berlin, Germany; 4Department of Neuroradiology, Charité—Universitätsmedizin Berlin, Corporate Member of Freie Universität Berlin, Humboldt-Universität zu Berlin, and Berlin Institute of Health, Augustenburger Platz 1, 13353 Berlin, Germany

**Keywords:** telemedicine, teleradiology, diagnostic performance, intracranial arterial aneurysm detection

## Abstract

**Purpose:** To evaluate a commercially available mobile device for the highly specialized task of detection of intracranial arterial aneurysm in telemedicine. **Methods:** Six radiologists with three different levels of experience retrospectively interpreted 60 computed tomography (CT) angiographies for the presence of intracranial arterial aneurysm, among them 30 cases with confirmed positive findings. Each radiologist reviewed the angiography datasets twice: once on a dedicated medical-grade workstation and on a commercially available mobile consumer-grade tablet with an interval of 3 months. Diagnostic performance, reading efficiency and subjective scorings including diagnostic confidence were analyzed and compared. **Results:** Diagnostic performance was comparable on both devices regardless of readers’ experience, and no significant differences in sensitivity (66–87.5%) and specificity (79.4–87%) were found. Results obtained with tablets and medical workstations were also comparable in terms of subjective assessment across all reader groups. **Conclusions:** There was no significant difference between tablet and workstation readings of angiography datasets for the presence of intracranial arterial aneurysm. Sensitivity, specificity, efficiency and subjective scorings were similar with the two devices for all three reader groups. While medical workstations are 10 times more expensive, tablets allow higher mobility especially for radiologists on call.

## 1. Introduction

Digitalization in healthcare confronts the radiology community with a rapid succession of technological developments and advances. Against this background, it has become a major challenge to benefit from the new techniques without incurring excessive costs. Therefore, radiological applications play a key role in implementing economic strategies in daily interdisciplinary workflow. One such measure is the establishment of a nationwide teleradiology network [1,2]. With such a network, only a few specialists are needed to efficiently provide high-level diagnostic services at several locations simultaneously, thus raising the standard of care also in rural regions [3]. Almost every university hospital in Germany now offers teleradiology services [3]. To ensure the highest possible quality even outside core working hours, these services are regularly confronted with the need to ensure image evaluation by neuroradiologists with interventional experience. This is the case, for example, when patients present with an acute subarachnoid hemorrhage and an underlying aneurysm requiring prompt intervention [4]. Experienced radiologists are consulted by telephone by the radiologists on site during night shifts or in emergency situations in order to enable rapid findings in time-critical situations and thus ensure rapid therapy (Figure 1).

There are two fundamental problems in the teleradiology setting. First, the radiologist making a remote diagnosis should have a medical-grade workstation that meets the specific requirements for medical use. Having such a workstation available at home is sometimes not possible [2]. Several nonmedical monitors have already been evaluated for their suitability for radiological applications [2,5,6,7,8,9,10]. Secondly, remote reading of image sets requires the transmission of large amounts of data, e.g., when reconstructed images need to be reviewed. This can significantly delay transmission on suboptimal digital networks—a common problem in rural areas. Portable tablets or other devices without medically calibrated screens are currently used by radiologists outside the hospital environment when prompt decision-making is required and no other option is available. In addition, nonmedical devices are increasingly used in other settings such as emergency care [11,12,13,14,15]. However, the display characteristics of such devices can degrade image quality and thus affect the radiologist’s diagnostic performance [16]. Therefore, several guidelines for medical monitors have been proposed to ensure appropriate grayscale differentiation to meet clinicians’ needs [17,18]. The DICOM grayscale standard display function (GSDF)—a mathematically defined luminance output relative to digital input—should be available to ensure an adequate grayscale range of displays for reading radiological images [2,17,18]. Liquid crystal display (LCD) technology is generally used for medical images [17]. However, new consumer-grade displays using light-emitting diode (LED) technology cannot be integrated into clinical practice because high requirements must be met to be approved for medical use. As medical displays are expensive, and high-quality consumer displays are widely available and can provide comparable image quality, several software packages are now available for effective grayscale calibration of consumer displays [2,5,9,10].

The aim of this study therefore was to compare the subjective image quality and resulting diagnostic accuracy of medical-grade digital screens with those of consumer displays in order to derive a practical approach to process optimization in the setting of telemedicine.

## 2. Materials and Methods

### 2.1. Study Design

This study was conducted retrospectively in a single center between September and December 2021. The study protocol conforms to the ethical guidelines of the 1975 Declaration of Helsinki. Patients who presented to the emergency department with suspected bleeding between June 2016 and January 2018 were retrospectively screened. Cases with a history of intracranial surgery were excluded and the first 30 cases with aneurysm detection and the first 30 cases with aneurysm exclusion based on initial clinical reports and confirmation by the head of the neuroradiology department with more than 25 years of experience in clinical CT reading were included in our retrospective analysis. The reference reading was performed using all possible (routinely generated) reconstructions on a medical-grade monitor workstation and patients’ medical history up to the time of their cranial CT scan (Figure 1). 

The resulting 60 CT datasets, including 30 cases with minimum one proven therapy-naive intracranial aneurysm, were retrospectively read by six radiologists with three different experience levels in reading computed tomography (CT) images: two experienced neuroradiology specialists (specialists) with 10 and 15 years of experience, two experienced body radiology consultants (consultants) with 10 and 12 years of experience, and two less experienced body radiology residents with 2 and 3 years of experience (residents) (Figure 2).

All cases were read twice by every reader. In the first round, one radiologist from each experience group randomly used either a calibrated medical-grade (workstation) grayscale screen (EIZO RadiForce RX250, EIZO, Hakusan, Japan) that met DICOM standards, or a consumer grade (tablet) mobile device (iPad Pro, Apple Inc., Cupertino, CA, USA). After an interval of 3 months, the second reading of all cases was performed with every reader using the device that was not used in the first round.

### 2.2. CT Imaging 

All patients included in our retrospective analysis were examined on 64-multislice CT scanners (Revolution HD and Revolution EVO, GE Healthcare, Milwaukee, WI, USA) after administration of a non-ionic contrast agent (either Ultravist 370, Bayer Schering, Berlin, Germany or Xenetix 350, Guerbet, Villepinte, France). The contrast agent was administered as a single 60 mL bolus at an injection rate of 4.0 mL/s, followed by a 20 mL saline flush using a mechanical injection system (Medtron CT2, MEDTRON AG, Saarbruecken, Germany). Automatic bolus tracking software (SmartPrep, GE Healthcare, Milwaukee, WI, USA) in a region of interest (ROI) placed in the aortic arch was used to determine the optimal time for angiography of the intracranial vessels (cut-off > 100 HU, approximately 20 s delay). Primary reconstruction was performed in axial orientation with 0.625 mm slice thickness, followed by creation of multiplanar reconstructions (MPRs) in sagittal and coronal planes and a maximum intensity projection (MIP) series.

### 2.3. Devices

The first round of reading per reader was performed randomly either on a workstation with medical grade monitors (EIZO RadiForce RX250, EIZO, Hakusan, Japan) calibrated according to German industry standards (DIN 6868-157) with a price (on 28 June 2022) of 2790.00 EUR. The 21.3″ Color LCD Monitor is a 2-megapixel high-brightness color monitor ideal for the accurate display of CT, MRI and X-ray grayscale images and color images such as 3D color rendering or nuclear medicine scans. For these screens, maximum luminance was 800 cd/m^2^. The contrast ratio, i.e., r′ = L′max/L′min (maximum contrast), was 1400:1. 

As second option, the images were displayed on a tablet (iPad Pro; Apple Inc., Cupertino, CA, USA) with a price (on 28 June 2022) of 692.00 EUR (Liquid Retina Display 12.9″ Multi-Touch Display (32.77 cm diagonal) with LED backlighting and in-plane switching (IPS) technology, resolution of 2732 × 2048 pixels at 264 ppi, ProMotion technology, wide color gamut display (P3), true tone display, anti-reflective coating, 1.8% reflectance, 600 nits brightness). The screen was used with the default consumer level settings and was not calibrated. After an interval of 3 months, the second reading of all cases was performed with every reader using the device that was not used in the first round. Images were displayed using Visage v. 7.1.10 (Visage Imaging GmbH, Berlin, Germany).

### 2.4. Reading Efficiency

Reading efficiency was measured as: Time to detection of first (clinical relevant) aneurysm orTime to completion of reading a dataset by confirming or ruling out further aneurysms.

Maximum reading time per case was 5 min. All readers were monitored by a blinded observer who measured reading time. 

Diagnostic accuracy and time to diagnosis regarding the first (clinical relevant) aneurysm were compared between the three experience levels of readers and the two types of reading devices using the results achieved by a neuroradiological specialist (head of department on workstation) as gold standard.

### 2.5. Subjective Image Quality

Subjective parameters were assessed on five-point Likert scales; results are presented as medians and their interquartile ranges (IQR) [18]. 

Image noise (5 = minimal image noise; 4 = less than average noise; 3 = average noise; 2 = above average noise; 1 = unacceptable image noise)Image contrast (5 = excellent; 4 = above average contrast; 3 = acceptable image contrast; 2 =suboptimal image contrast; 1 = poor image contrast)Conspicuity of small structures (5 = excellent; 4 = above average; 3 = acceptable; 2 = suboptimal; 1 = unacceptable)Visibility of aneurysm(s) (if present) (5 = excellent; 4 =above average; 3 = acceptable; 2 =suboptimal; 1 = poor)Artifacts (5 = no artifacts; 4 = minor artifacts not interfering with diagnostic decision making; 3 = minor artifacts affecting visualization of minor structures but diagnosis still possible; 2 = major artifacts affecting visualization of major structures but diagnosis still possible; 1 = very severe artifacts precluding diagnosis)Diagnostic confidence level (5 = very confident; 4 = confident; 3 = confident only for a limited clinical entity; 2 = unconfident; 1 = very unconfident)

### 2.6. Statistics

Statistical analysis was performed using Stata IC15 (StataCorp, 2017, College Station, TX, USA). All values, unless otherwise stated, are given as mean and standard deviation (SD). The Wilcoxon signed-rank test was used to compare mean reading times between readers, rounds and sessions. Furthermore, stepwise forward linear modeling was performed for correlations between reading times, readers, sessions and hardware. Sensitivity (Se) and specificity (Sp) were determined as measures of diagnostic performance. McNemar’s test was conducted to compare Se and Sp between workstation and tablet readings. Intrarater agreement between devices was assessed using percentage of agreement and kappa statistic. A *p*-value less than 0.05 was considered statistically significant except for reading times, which were compared using the Wilcoxon signed-rank test with a significance level of 0.016 (0.05/3 reader groups after Bonferroni’s adjustment).

## 3. Results

### 3.1. Study Population

The study population consisted of 60 patients (female= 53% [32/60], male= 47% [28/60]; mean age of 68 (SD = 16) years; age range = 25–94 years). The mean long axis of the largest aneurysm within the 30 cases with positive finding of which 10 had more than one aneurysm was 4.8 mm (SD 2.8, range = 2–14 mm) and the mean short axis was 3.4 mm (SD 1.4, range =1–8 mm) in reference reading. 

In terms of diagnostic confidence between readers and devices, interreader agreement was 90.8% (kappa 0.813) for consultants, 94.9% (kappa 0.896) for specialists and 89% (kappa 0.753) for residents. 

### 3.2. Subsection Diagnostic Accuracy

Across all three levels of reader experience, Se and Sp did not differ significantly between tablet and workstation readings. There was no significant difference in reading performance between the devices (Figure 3).

### 3.3. Reading Efficiency

For each examination, readers were given a maximum of 5 min for a final assessment, set as the number of aneurysms. Here, the time to find the first aneurysm and to the overall assessment was to be stopped.

Reading times did not differ significantly between tablet and workstation in all experience groups (Figure 4).

First aneurysm detection in all reader groups and on all devices was made in less than one minute. 

Final assessment target of 5 min for the evaluation of further aneurysms was not exceeded in the experience groups or both devices.

### 3.4. Subjective Parameters

Image noise was rated as minimal (median score of 5) by consultants on both the workstation and the tablet (*p* = 0.006). Specialists rated noise as average on the workstation (median score of 3) and as above average on the tablet (median score of 2) (*p* < 0.001). Residents classified noise as less than average on both devices (median scores of 4) (*p* = 0.353) (Figure 5).

Image contrast was rated to be excellent by consultants (median score 5) both on the workstation and on the tablet (*p* = 0.002).

The specialists rated both workstation and tablet as acceptable for image contrast (median rating 3) (*p* = 0.369). The residents rated both the workstation and the tablet as above average for image contrast (median rating 4) (*p* = 0.249) (Figure 5).

The visualization of small structures was rated by consultants as excellent visualization (median score 5) both on the workstation and on the tablet (*p* < 0.001).

The specialists rated both workstation and tablet as acceptable for image visibility (median score 3) (*p* = 0.425). The residents rated both the workstation and tablet as above average visibility (median score 4) (*p* = 0.824) (Figure 5).

The visualization of aneurysms was rated by consultants as excellent (median rating 5) both on the workstation and on the tablet (*p* = 0.009).

The specialists rated both the workstation and the tablet as having acceptable visibility (median score 3) (*p* = 0.083). The residents rated both the workstation and the tablet with above average visibility (median rating 4) (*p* = 0.087) (Figure 5).

The artifact load was rated as non-existent (median score 5) by consultants on both the workstation and the tablet (*p* = 0.724).

The specialists rated the workstation as well as the tablet with minor artifacts affecting visualization of minor structures but diagnosis as still possible (rating 3) (*p* = 0.417). The residents rated both the workstation and the tablet as minor artifacts not interfering with diagnostic decision making (median score 4) (*p* = 0.211) (Figure 5).

The potential diagnostic confidence was assessed by consultants as probable (median score 4) for the tablet and as completely confident (median score 5) for the workstation (*p* < 0.001).

The specialists and the residents rated the workstation and the tablet as probable confident (rating 4) (Figure 5). 

## 4. Discussion

Our study shows that diagnostic accuracy in the detection and exclusion of intracranial aneurysms and reading efficiency (time to diagnosis) do not differ significantly between medical workstations and consumer-grade tablets. The use of a tablet does not appear to affect the diagnostic confidence of radiologists with different levels of experience or the assessment of artifacts and image noise.

The advent of new generations of high-performance digital displays has raised the question of whether expensive medical displays are still necessary to ensure optimal diagnostic accuracy when reading radiological images [1,2]. Several studies have investigated the use of nonmedical displays such as tablets especially in terms of diagnostic accuracy [2,10,11,12,13,14,15,18,19,20]. Our results for diagnostic accuracy confirm these previous studies and showed no significant difference between the devices. As diagnostic performance may depend on experience when new equipment is used, we also analyzed the results separately for specialists, consultants and residents. In this subgroup analysis, the level of experience did not affect diagnostic accuracy.

Similar to previous studies comparing different mobile devices [10,13] with workstations, we found the tablet to be suitable for the investigated setting with CT-images.

Different results were reported by Krupinski, who investigated the diagnostic accuracy of radiologists viewing clinical X-ray images on a high-quality medical display compared to a high-quality commercial color display. The radiologists reviewed a set of 50 digital chest radiographs for the detection of pulmonary nodules, and the investigators found significantly higher diagnostic accuracy for medical-grade displays [21]. 

Medical-grade displays can be expected to provide an advantage when radiologists need to detect low-contrast objects, which requires maximum display performance [2,21,22,23]. Most studies published to date provide only quantitative accuracy data, while we also tried to include subjective impressions of our readers, which they rated on Likert scales.

Most importantly, we found that, in all reader experience levels, the subjective diagnostic confidence was not significantly different between the two display types, which is critical in an emergency situation. Subjective ratings of artifacts, image noise, image contrast and visibility of aneurysms or small structures also showed no significant differences between medical-grade and non-medical grade equipment. We therefore conclude that recent consumer grade displays are comparable to medical-grade displays in terms of subjective image quality.

The major limitation of our study is that all readers reviewed the 60 study datasets twice to compare medical-grade and non-medical-grade displays. Although there was a time interval of 3 months between the two readings, we cannot completely rule out a recall bias. Second, although blinded to the clinical data, all readers were aware of the study design, which may have introduced some diagnostic bias. 

Future studies should evaluate even smaller devices (e.g., smartphones) for telemedical reporting in 5-G networks.

In conclusion, the use of a new generation of tablets does not reduce diagnostic accuracy in detecting and ruling out intracranial aneurysms in CT scans compared to medical-grade displays. Subjective image quality is also comparable, so we conclude that these tablets may be suitable for reading CT datasets, including structures with the shape and size of intracranial aneurysms. This may have an economic and practical impact by reducing costs and increasing time effectiveness, especially regarding a wider availability of teleradiology services. The results could simplify daily radiology practice and have an economic impact as consumer grade devices are less expensive, widely available and mobile.

## Figures and Tables

**Figure 1 diagnostics-12-02461-f001:**
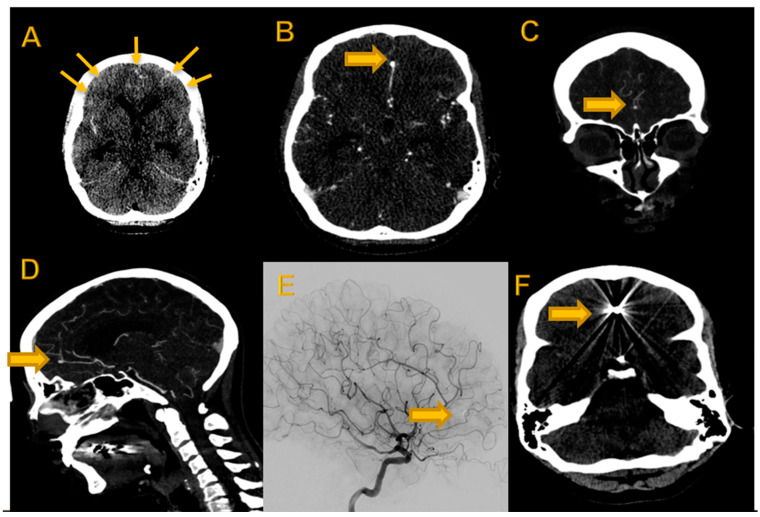
Imaging dataset of 42-year-old female patient with acute intracranial hemorrhage due to ACA aneurysm. (**A**) Non-contrast-enhanced axial cCT showing SAH [arrows]. (**B**) Contrast-enhanced axial slice with ACA aneurysm [block arrow]; (**C**) coronal reconstruction MIP with ACA aneurysm (block arrow); (**D**) sagittal reconstruction MIP with ACA aneurysm (block arrow) and (**E**) DSA after clipping (block arrow). (**F**) Postinterventional non-enhanced cCT with clip (block arrow).

**Figure 2 diagnostics-12-02461-f002:**
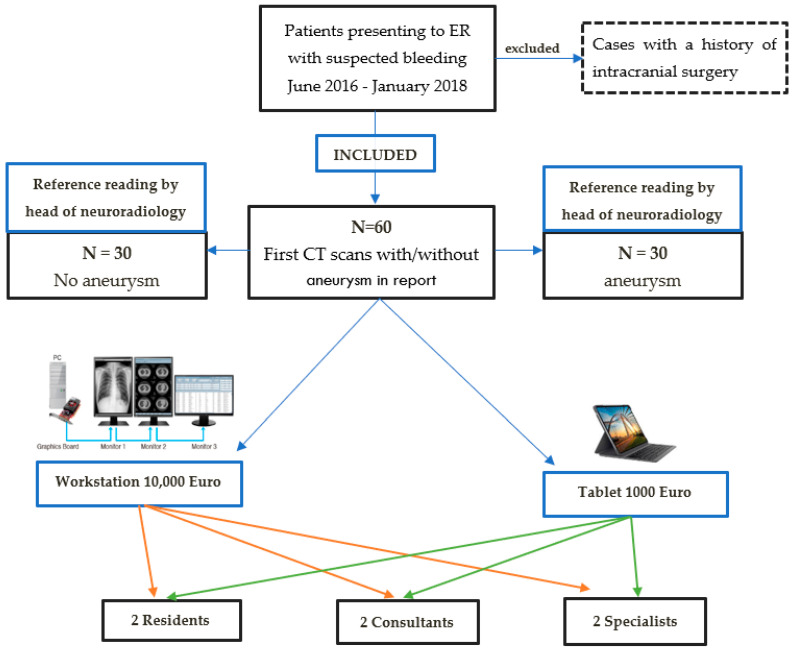
Study design.

**Figure 3 diagnostics-12-02461-f003:**
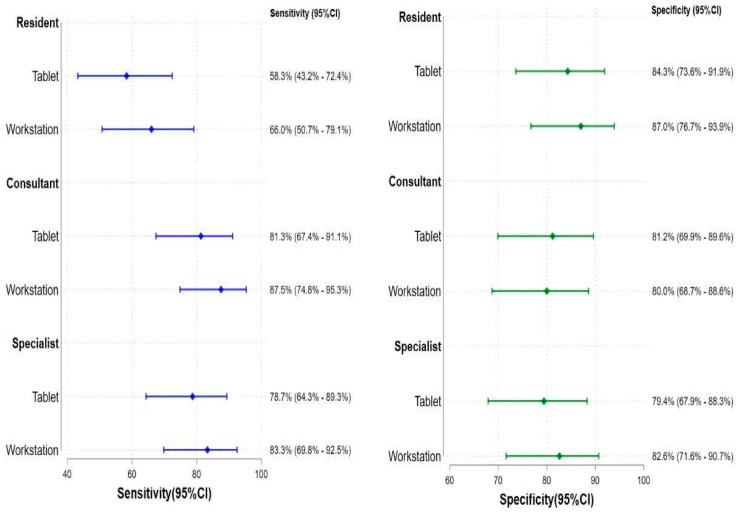
Diagnostic performances between Workstation and Tablet. *p*-value was performed using McNemar’s test for comparing Se and Sp between workstation and tablet.

**Figure 4 diagnostics-12-02461-f004:**
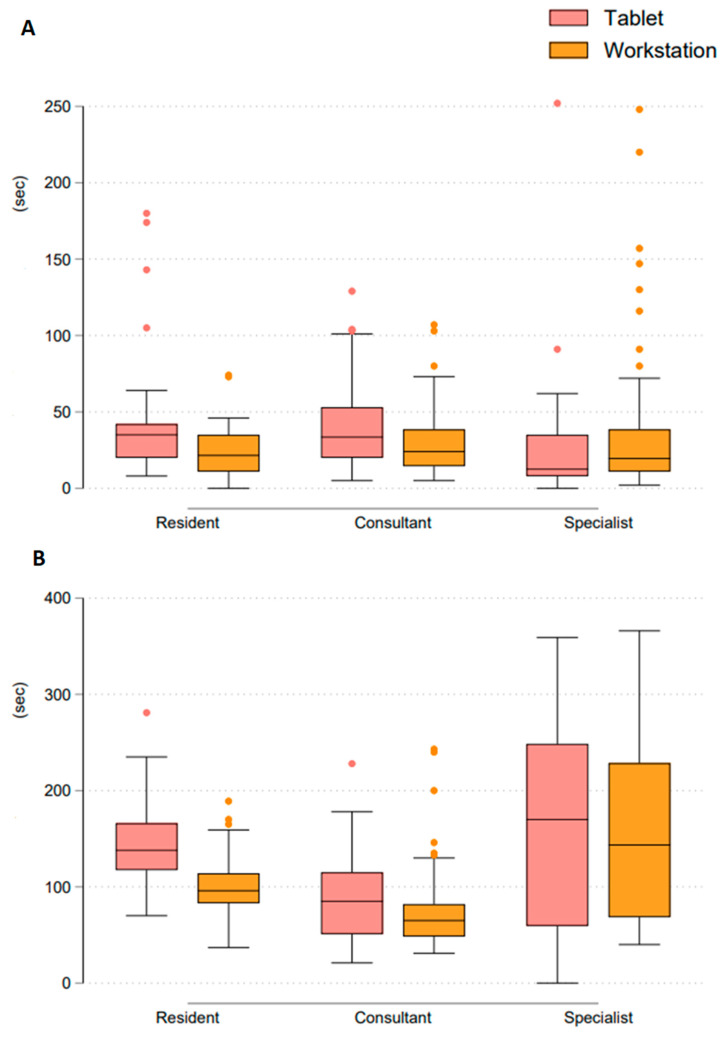
(**A**) Comparison time to first aneurysm detection n = number of cases which was detected to have an aneurysm. (**B**) Comparison time to confirm/rule out further aneurysms (overall time).

**Figure 5 diagnostics-12-02461-f005:**
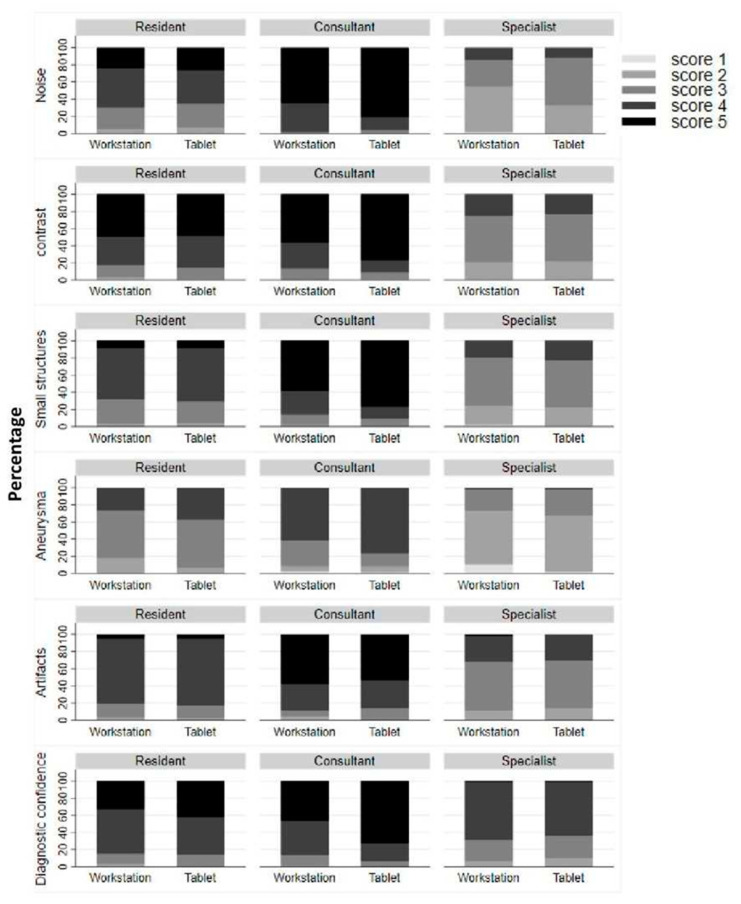
Results for subjective image quality parameters. Please find details in the methods section.

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
