# Peer review of "Tablets as an Option for Telemedicine—Evaluation of Diagnostic Performance and Efficiency in Intracranial Arterial Aneurysm Detection"

_diagnostics, 2022, doi:10.3390/diagnostics12102461_

Round 1

Reviewer 1 Report

In my opinion this paper is well written and really useful for its practical implications. Simply note that figure 2 has been wrongly indicated as Fig. 1, please correct.

Author Response

Dear Reviewer,

Thank you very much for your helpful recommendations for improving our manuscript.

In revising our manuscript, we have dealt with your suggestions.

The numbering of the figure has been corrected (page 3).

For your convenience, the changes are highlighted in the tracked changes mode in the revised manuscript.

We hope that our revised manuscript is now deemed suitable for publication in your journal.

Please feel free to make any further suggestions.

Sincerely yours,

Elif Can

Georg Böning

Reviewer 2 Report

Thank you for the opportunity to review this paper by E. Can and colleagues.

The paper aims to establish the differences between tablet and workstation readings of angiography datasets for the presence of intracranial arterial aneurysm. The results demonstrated that the diagnostic performance was comparable on both devices no matter what is the readers’ experience and there are not found significant differences in sensitivity and specificity.

The paper is well written. The language used is concise, the methodological approach and findings are clearly described. The structure of the paper has a clear and logical flow. Some limitations of the research are underlined.

I have some concerns/improvement suggestions:

1. Fig 1 has to be inserted closer to its first reference;

2. In page 4, the figure is Fig 2 (not Fig 1, as written under it);

3. In “Devices” (from page 5), the time of first reading is not correct:

a.       First paragrapf: “The first round of reading per reader was performed randomly either on a workstation with medical grade monitors (EIZO RadiForce RX250, EIZO, Hakusan, Japan) calibrated according to German industry standards (DIN 6868-157) with a price of 2,790.00 EUR on 28th June 2022”;

b.       Second paragraph: “As second option, the images were displayed on a tablet (iPad Pro; Apple Inc., Cupertino, CA, USA) with a price of 692 EUR on 28th June 2022”;

c.       Second paragraph: “After an interval of 3 months, the second reading of all cases was performed with every reader using the device that was not used in the first round”

because the interval of 3 months after 28th June 2022 is after the date if this revision;

4. The quality of Fig 3 has to be improved;

5. Among the 20 references used by the authors, only 4 are to papers published in the last 5 years and there is no reference from the last 2 years, although the technology has evolved spectacularly in the last period. An update of the references would be welcome;

6. No future research opportunities are suggested in the Discussion section. Please, mention if there is any.

For these reasons, I would recommend the revision of the manuscript.

I hope my feedback is useful to the authors in improving the paper and wish them all the best in pursuing this important area of research.
